# The dilemma of physical activity questionnaires: Fitter people are less prone to over reporting

**Kaja Meh** [1]*, **Vedrana Sember**[1], **Maroje Sorić**[1,2], **Henri Vähä-Ypyä** [3], **Paulo Rocha** [4], **Gregor Jurak**[1]

1 Faculty of Sport, University of Ljubljana, Ljubljana, Slovenia, 2 Faculty of Kinesiology, University of Zagreb, Zagreb, Croatia, 3 UKK-Institute, Tampere, Finland, 4 Portuguese Institute of Sport and Youth, Lisbon, Portugal

* kaja.meh@fsp.uni-lj.si

**Data Availability Statement:** All relevant data are within the paper and its Supporting Information files.

**Funding:** This research was co-funded by the Erasmus+ Programme of the European Union

## Abstract

Physical activity questionnaires (PAQs) are a popular method of monitoring physical activity, although their validity is usually low. Descriptions of physical activity levels in questionnaires usually rely on physical responses to physical activity. Therefore, we hypothesised that the validity of PAQs would be higher in the more physically fit group of participants. To test this, we conducted a validation study with 179 adults whom we divided into three fitness groups based on their cardiovascular fitness and age. Participants were measured for one week using the UKK RM42 accelerometer and self-reported their physical activity using IPAQ-SF, GPAQ, and EHIS-PAQ. We analysed the differences between fitness groups in terms of validity for each PAQ using ANOVA. We also performed an equivalence testing to compare the data obtained with the PAQs and the accelerometers. The results showed a significant trend toward higher validity for moderate to vigorous physical activity from the low to high fitness group as assessed by GPAQ and IPAQ-SF (low, intermediate and high fitness group: 0.06–0.21; 0.26–0.29; 0.40, respectively). The equivalence testing showed that all fitness groups overestimated their physical activity and underestimated their sedentary behaviour, with the high fitness group overestimating their physical activity the least. However, EHIS-PAQ was found to agree best with accelerometer data in assessing moderate to vigorous physical activity, regardless of fitness group, and had a validity greater than 0.4 for all fitness groups. In conclusion, we confirmed that when using PAQs describing physical responses to physical activity, participants' fitness should be considered in the interpretation, especially when comparing results internationally.

## Introduction

Physical inactivity significantly impairs health [1–3] and is increasingly becoming a burden in developed countries [3, 4]. The COVID -19 pandemic fostered this trend [5, 6] because movement restrictions were put in place to contain the spread of the virus. However, the long-term

within the project EUPASMOS No. 590662-EPP-1-2017-1-PT-SPO-SCP, and Slovenian Research Agency within the Research programme Bio-psycho-social context of kinesiology No. P5-0142. The funders had no role in study design, data collection and analysis, decision to publish, or preparation of the manuscript.

**Competing interests:** I have read the journal's policy and the authors of this manuscript have the following competing interests: Author Henri Vähä-Ypyä is employee of the UKK Institute which is the main owner of company UKK tervepalvelut that developed RM42 accelerometers. This does not alter our adherence to PLOS ONE policies on sharing data and materials

consequences of isolation and social distancing on behavioural patterns are unknown [7]. Indeed, a combination of movement behaviours across the day is very important for health outcomes because it predicts health risk better than a single behaviour [8, 9]. Therefore, the 24-hour movement behaviour paradigm, which combines three behaviours (physical activity, sleep, and sedentary behaviour) within 24 hours, is a hot research topic. A recent meta-analysis showed that shifting from undesirable physical activity behaviours (sedentary behaviour) to physical activity (PA) is associated with several health benefits such as lower body mass index (BMI) and mortality [10].

All three movement behaviours can be assessed with more objective measurements, like accelerometers, and subjective measurements, like questionnaires or diaries. Accelerometers are considered to be more valid measures of PA than self-reports [11]. However, accelerometers measures depend on movement of certain parts of body (e.g., hip, wrist) and their metrics (movement counts, bodily position etc.) [12], which could not detect all habitual movements. Consequently, validity between different types of accelerometers varies [13]. Because of feasibility of performing large scale studies and the above-mentioned characteristics of accelerometers, PA questionnaires (PAQs) are still an important part of PA research. They provide individuals perception of PA and in combination with accelerometers and other measurement devices provide richer data, needed for understanding of human behaviour [14]. However, they should be validated to obtain reliable and valid results. Despite described weaknesses, accelerometers are the best instruments to assess their validity, since they can measure habitual movement behaviour with movement sensor. Moreover, comparison of PAQs with doubly labelled water as a golden standard for measuring PA showed low correlations between the two methods [15, 16] and systematic bias in underestimation of energy expenditure [17]. Oposite, comparison of accelerometers with doubly labelled water demonstrate high correlations [18, 19].

The most commonly used PAQs in population-based studies are the International physical activity questionnaire-short form (IPAQ-SF) [20], the Global physical activity questionnaire (GPAQ) [21], and the PAQ from the European health interview survey (EHIS-PAQ) [22]. All three questionnaires assess PA and sedentary behaviour, but not sleep. The descriptions are provided to better understand the questions included in the PAQs and with the intention of distinguishing between PA of different intensities. The descriptions in IPAQ-SF and GPAQ rely on physical responses to PA and use explanations of heavy breathing and increased heartbeat to distinguish between moderate (MPA) and vigorous PA (VPA). For example, the GPAQ uses the following description of VPA: "Do you do any vigorous-intensity sports, fitness, or recreational (leisure) activities that cause a large increase in breathing or heart rate like [running or football], for at least 10 minutes continuously?" and MPA: "Do you do any moderate-intensity sports, fitness, or recreational (leisure) activities that cause a small increase in breathing or heart rate, such as brisk walking, [cycling, swimming, volleyball] for at least 10 minutes continuously?" Both descriptions are highly subjective, as participants may perceive the physical signs of PA differently. In addition, less fit participants perceive heavy breathing or increased heart rate at a lower PA intensity than fitter individuals. Inexperienced and inactive participants may not know what a sharp increase in heart rate or breathing is and may interpret even the slightest changes as VPA. Even everyday activities, such as climbing stairs, may elicit different physical responses in less and very fit participants and consequently lead to different responses to the same PA question. All this could lead to associating the measurement error of the PA questionnaires with the fitness level of a respondent. On the other hand, EHIS-PAQ is not based on the description of physiological responses, but focuses on the description of activities, e.g.: "In a typical week, on how many days do you carry out sports, fitness or recreational (leisure) activities for at least 10 minutes continuously?".

A study by Fogelholm and colleagues [23] found differences in self-reported PA and cardio-respiratory fitness between inactive (divided into two groups) and active participants (divided into three groups). Cardiorespiratory fitness increased from the least active group to the more active groups. They also reported an unusual phenomenon. The most physically active group (based on the health enhancing PA from the IPAQ), was the 'overreporters' group; older participants with low physical fitness and more abdominal obesity who overreported their PA in the IPAQ, but had similar fitness levels to the lowest 20% by IPAQ grouping. Considering the differences in PA self-report, the validity of PAQs might differ between different groups of participants. There are few studies that have compared the validity of PAQs in different fitness groups, and these generally showed differences in criterion validity between fitness groups. In the Active Australia Survey (ActiGraph GT3X accelerometer was used as an objective measure of PA), lower criterion validity was found for moderate to vigorous PA (MVPA; Spearman $\rho$ = 0.165 and Spearman $\rho$ = 0.192) in participants with overweight and obesity compared to the healthy weight group (Spearman $\rho$ = 0.361) [24]. Comparison of fit and unfit participants in the Energy Balance Study performed by SenseWear accelerometers [25] found that both fit and unfit participants overestimated VPA, but unfit participants overestimated their VPA by more than 600%, whereas fit participants overestimated their PA by less than 300%. Both groups underestimated sitting time, while fit participants underestimated MPA.

Because previous studies showed some differences in self-reported PA between differently fit individuals, we aimed to analyse this problem by comparing the criterion validity of the most commonly used adult PAQs for adults between groups of individuals with different fitness levels. We hypothesised that the validity of all PAQs used would be higher in groups of participants with higher fitness.

## Materials and methods

### Study design and participants

Participants in the study were gathered through 9 Slovenian primary schools using snowball sampling. Parents, grandparents, and adult siblings of 12- to 14-year-old students were invited to participate in the study. Only participants whose PA was not affected by a health condition were included. A kinesiologist reviewed the participants' health status and decided whether they could participate in the study. Permission to conduct the study was granted by the Ethical Committee of the Faculty of Sport in Ljubljana in accordance with the Declaration of Helsinki (No: 6:2020–274). The data of the present study were obtained within the European project EUPASMOS No. 590662-EPP-1-2017-1- PT-SPO-SCP.

A total of 399 participants volunteered for the study (41% male, mean age = 41, SD = 14, mean BMI = 25, SD = 4). We excluded 220 individuals due to an incomplete study protocol (invalid questionnaire and/or accelerometer data; only participants who completed all three PAQs were included in the study) or missing physical fitness data, leaving 179 participants (42% male, mean age = 47, SD = 10; mean BMI = 25, SD = 5) included in the analysis. While the gender distribution of our sample was similar to the initial sample, excluded participants were approximately 10 years younger than those included in the analysis. More importantly, there were no differences in BMI or accelerometer-measured MVPA between these two groups. Participants attended the study for a full week. At the first visit, all participants were provided with accelerometers and familiarized with their use. They were instructed to wear the accelerometers for seven consecutive days (24 hours/day), except during water activities (e.g., swimming, showering, sauna visits). After seven days, all participants returned for the second visit. We collected their accelerometers and participants continued with anthropometric

measurements (height, weight, waist circumference) and physical fitness testing. Participants then completed the three selected PAQs in a random order.

## Subjective measures of PA

Three adult PAQs most commonly used in the European Union were used to assess PA: IPAQ-SF, GPAQ, and EHIS-PAQ. IPAQ-SF and GPAQ are standardised instruments and have been used for many years in different cultural settings [26, 27]. Both assess moderate and vigorous PA, transport PA (walking in IPAQ-SF) and sedentary behaviour. In addition, EHIS-PAQ item MV Aerobic Recreational Activity with additional walking and cycling was used as MVPA because EHIS-PAQ was not designed to measure total PA or MVPA [22]. The GPAQ is more detailed and includes separate questions for work PA and leisure time PA. EHIS-PAQ is part of the European Health Interview Survey and is used in all European Union Member States. The EHIS-PAQ does not measure the intensity of PA, but measures PA in areas relevant to public health, such as work, transport, and leisure domain [22]. All three questionnaires measure duration of PA and sedentary behaviour in minutes; with IPAQ-SF participants report their PA in the last week, while GPAQ and EHIS-PAQ ask about PA in a regular week. On all three PAQs, participants self-reported number of days in each activity and daily time spent in the activity. From that we calculated the weekly PA of the participants using the original scoring protocol. Sedentary behaviour (minutes per day) and moderate to vigorous recreational activity from EHIS-PAQ (minutes per week) were already self-reported in the units presented in the paper. Participants in our study completed the Slovenian versions of PAQs, that were translated using the forward-backward translation method [28]. Two independent translators interpreted the PAQs from English into Slovenian and two other independent translators back into English. Then, the two English versions were compared, and we decided on the best translation. The participants completed online form of the selected PAQs, all during the same visit in a randomised order. The reliability and validity of all three PAQs have already been tested, but mostly on English versions; IPAQ-SF and GPAQ have already been validated in many EU countries [26, 27], while the measurement characteristics of EHIS-PAQ have been tested, but not in all European Union countries [29].

Their reliability has been shown to be moderate to high (IPAQ-SF: Spearman's $\rho$ = 0.66 to 0.87 for PA and Spearman's $\rho$ = 0.50 to 0.95 for sedentary behaviour [27]; GPAQ: Spearman's $\rho$ = 0.67 to 0.73 for PA and Spearman's $\rho$ = 0.68 to 0.73 for sedentary behaviour [26]; EHIS-PAQ: ICC = 0.51 to 0.73 for PA [29]). Criterion validity of all three PAQs tested with the ActiGraph accelerometer was low for both PA (IPAQ-SF: Spearman's $\rho$ = 0.17 to 0.49 [30, 31]; GPAQ: Spearman's $\rho$ = 0.24 to 0.48 [32, 33]; EHIS-PAQ: Spearman's $\rho$ = 0.13 to 0.37 [29]) as well as for sedentary behaviour (IPAQ-SF: Pearson's $\rho$ = 0.16 to Spearman's $\rho$ = 0.28 [30, 34]; GPAQ: Spearman's $\rho$ = 0.19 to 0.42 [32, 33].

## Objective measures of PA

PA was measured using an RM42 triaxial accelerometer (UKK Terveyspalvelut Oy, Tampere, Finland). The accelerometer was worn on the right hip during waking hours and on the non-dominant wrist during sleeping hours. Acceleration data were acquired in a range of ± 16 G at a sampling rate of 100 Hz and stored on a hard disc for further analysis. The analysis of PA was based on the mean amplitude deviation (MAD) in six-second epochs [35]. MAD has been shown to be a valid indicator of incident oxygen consumption during locomotion [36]. For each epoch, MAD values were converted to METs (3.5 mL/kg/min oxygen consumption). The epoch-wise MET values were further smoothed by calculating an exponential moving average

for each epoch time point [37]. The smoothed data were analysed in 6-s epochs and the PA cut points were set as follows: 3.0 METs ≤ MPA < 6.0 METs and VPA ≥ 6.0 METs.

Sedentary behaviour (sitting and lying) and standing were identified for epochs where in which the predicted MET value was less than 1.5. The orientation of the accelerometer with respect to the gravity vector was taken as the reference, and the angle for posture (APE) estimation was determined from the orientation of the accelerometer with respect to the reference vector [38]. Body posture was classified as standing if the angle for body posture was less than 11.6˚, sitting if the angle for body posture was between 11.6˚ and 72.0˚, and lying if the angle for body posture was greater than 72.0˚. The epochal six-second values for posture were also smoothed by a one-minute exponential moving average.

A valid day was defined as one in which the monitor was worn for at least 600 minutes during awake time. Participants were required to wear the accelerometer for at least 4 valid days, one of which had to be a weekend day, to be included in the analyse.

## Anthropometry and physical fitness

Height (to the nearest 0.1 cm) and weight (to the nearest 0.1 kg) were measured using a Seca 799 electronic scale (Seca Germany, Hamburg, Germany), waist circumference was measured with measuring tape to the nearest 0.1 cm midway between the lowest rib and the iliac crest. We calculated body mass index (BMI) from height and weight. Participants were barefoot and wore light sports clothing during measurements, they were asked to wear athletic footwear during the 6-minute walk test. Participants' fitness level was determined using the 6-minute walk test [39], one of the most popular cardiorespiratory fitness tests for adults [40]. The test has been validated in healthy adult populations and can be used as valid test for assessing cardiorespiratory fitness [41, 42]. The test was performed in the school gym: A 30-m oval track was prepared for the participants. Cones were placed 5 meters apart to mark the track. Participants were familiarized with the test beforehand: they were first informed about the duration and aim of the test, and next the test protocol was demonstrated. Participants started the test at one of the cones and walked for 6 minutes. After each elapsed minute, they were informed how much time was left. After 6 minutes, they stopped, and the distance was measured to the nearest 1 m so that the number of full laps was counted and the remained distance from the starting point to finish was measured. Maximum of 4 participants performed the test at the same time. Each participant completed the test once.

## Statistical analysis

Data analysis was performed using IBM SPSS 27 software (Armonk, NY: IBM Corp), Microsoft Excel, and Jamovi [43, 44]. Fitness groups were formed by first dividing female and male participants separately into 4 age groups (18–34.99, 35–49.99, 50–64.99, and > 65 years). Second, we divided participants in each age group into terciles based on their 6-minute walk test distance. The 6-minute walk distance in the low fitness group was 461–630 m for males and 360–640 m for females, in the intermediate fitness group 528–690 m for males and 525–690 m for females and in the high fitness group 660–870 m for males and 585–840 m for females. Normality of the data was tested with the Kolmogorov-Smirnov test. Differences between groups were calculated with the Kruskal-Wallis test for nonnormally distributed data and ANOVA for normally distributed data using the Bonferroni correction with the Kruskal-Wallis test. Criterion validity was assessed with Spearman correlation coefficients. Validity values were categorised as follows: ≤0.29 very low, 0.30–0.49 low, 0.50–0.69 moderate, 0.70–0.89 high, and above 0.90 very high validity [45]. In addition, equivalence testing was conducted to evaluate the agreement between each PAQ and the accelerometer in assessing the duration of

MVPA and sedentary behaviour. The Confidence Interval Method [46, 47] was used to provide empirical evidence of equivalence between the selected measurements. Because the accelerometer data were used as a known reference value, we set bounds as raw values and defined them as ± 15% [46]. Therefore, equivalence bounds for sedentary behaviour were set at ± 78.5 min/day and ± 58 min/week for MVPA.

## Results

Baseline characteristics of participants, stratified by fitness level, are shown in Table 1. There were no age differences among the three fitness groups, but the low fitness group had statistically higher mean BMI and waist circumference than the high fitness group. In addition, participants in the high fitness group reported more sedentary behaviour and less MPA and MVPA (except EHIS-PAQ moderate to vigorous recreational activity) compared to the other two groups. At the same time, UKK RM42 recorded the highest amount of MPA and MVPA in the high fitness group. Sedentary behaviour measured by accelerometer was similar in all fitness groups.

To compare criterion validity between fitness groups, we calculated Spearman's correlation coefficients for each PAQ (Table 2). We found statistically significant correlations for sedentary behaviour in all three fitness groups and for all PAQs. Nevertheless, criterion validity was low to moderate for all PAQs and in all fitness groups. Validity results for the GPAQ were similar in all fitness groups, while the intermediate fitness group showed higher validity results for the IPAQ-SF and EHIS-PAQ.

For MVPA, validity was lower in the low fitness group for IPAQ-SF and especially for GPAQ. However, for EHIS-PAQ, validity was slightly lower in the high fitness group than in the intermediate and low fitness group. The validity of MPA showed similar patterns in all groups, while the validity of VPA was very low in all fitness groups and showed no statistically significant correlations.

To assess the agreement between self-reported and accelerometer-measured PA and sedentary behaviour in the three fitness groups, we created Bland-Altman plots for each PAQ and fitness group (Figs 1 and 2). There were differences in PA and sedentary behaviour duration between accelerometer and PAQs in all fitness groups. The duration of self-reported PA was longer compared to accelerometer and sedentary behaviour duration was shorter when using PAQs. The differences between the accelerometer and PAQs for sedentary behaviour were smallest for the high fitness group for IPAQ-SF and GPAQ, while for EHIS-PAQ the smallest differences were found in the low fitness group. The difference in duration of sedentary behaviour was largest for participants from intermediate fitness group for all three PAQs. The differences in MPA and MVPA duration between PAQs and accelerometer UKK RM42 were the lowest in high fitness group. These results were also influenced by outliers in all three groups, as shown in Fig 2. The limits of agreement differed between groups; the limits were tightest for the high fitness group for PA and sedentary behaviour, suggesting that the high fitness group's results were most equivalent to the accelerometer. On the other hand, the limits of agreement were greatest in the intermediate fitness group, where the bias between the two measurements was also greatest, especially for the PA. There were quite a few outliers in the high and intermediate fitness group for IPAQ-SF and GPAQ. The outliers show a substantial difference between the accelerometer and PAQ in a few individuals.

The results of the equivalence testing are shown in Figs 3 and 4. In the equivalence testing Two One-Sided Tests were used; we performed two paired-samples T-tests: for sedentary behaviour and for MVPA. The differences between the accelerometer UKK RM 42 and the PAQs were statistically significant for sedentary behaviour (IPAQ-SF: $t(171)$ = -12.6,

**Table 1. Baseline characteristics of participants across three fitness groups (data shown are median and (interquartile range)).**

| | Fitness group | | | | | | | | |
| --- | --- | --- | --- | --- | --- | --- | --- | --- | --- |
| | Low fitness | | | Intermediate fitness | | | High fitness | | |
| | Male (N = 16) | Female (N = 36) | Total (N = 52) | Male (N = 24) | Female (N = 38) | Total (N = 62) | Male (N = 35) | Female (N = 30) | Total (N = 65) |
| Age (years) | 43.0 (9.0) | 44.0 (8.5) | 43.0 (7.8) | 44.5 (7.0) | 44.0 (13.0) | 44.0 (10.0) | 45.0 (7.0) | 43.0 (9.5) | 44.0 (9.0) |
| BMI (kg/m$^2$) | 27.5 (5.7) | 25.1 (10.0) | 26.4 (9.0) | 26.7 (4.8) | 24.3 (5.8) | 25.7 (6.0) | 24.5 (4.7) | 22.5 (3.3) | 23.4 (5.3) |
| Waist circumference (cm) | 103.0 (20.7) | 89.5 (24.5) | 93.1 (23.0) | 97.0 (14.0) | 84.0 (19.1) | 89.5 (17.6) | 90.0 (13.4) | 80.5 (13.5) | 85.0 (15.2) |
| 6-minute walk test (distance in m) | 547.5 (110.0) | 577.5 (90.0) | 570 (90.0) | 660.0 (43.0) | 660.0 (49.0) | 660 (49.0) | 750.0 (80.0) | 727.5 (86.0) | 750.0 (90.0) |
| Sedentary behaviour | | | | | | | | | |
| RM42 (min/day) | 500.7 (137.8) | 504.1 (179.0) | 502.8 (168.6) | 574.1 (207.7) | 501.7 (141.5) | 514.2 (145.9) | 534.1 (152.8) | 480.7 (150.8) | 496.0 (150.2) |
| IPAQ-SF (min/day) | 245.0 (142.5) | 450.0 (318.8) | 360.0 (281.3) | 287.5 (450.0) | 345.0 (270.0) | 317.5 (312.5) | 360.0 (240.0) | 420.0 (270.0) | 390.0 (240.0) |
| GPAQ (min/day) | 305.0 (270.0) | 480.0 (285.0) | 335.0 (292.5) | 240.0 (270.0) | 285.0 (337.5) | 270.0 (300.0) | 480.0 (360.0) | 360.0 (277.5) | 420.0 (300.0) |
| EHIS-PAQ (min/day) | 360.0 (330.0) | 420.0 (330.0) | 420.0 (330.0) | 300.0 (360.0) | 360.0 (360.0) | 300.0 (360.0) | 4200.0 (240.0) | 360.0 (360.0) | 420.0 (360.0) |
| Moderate physical activity | | | | | | | | | |
| RM42 (min/week) | 357.2 (349.6) | 303.8 (190.5) | 317.6 (199.5) | 424.1 (296.6) | 292.9 (281.3) | 331.5 (295.6) | 383.6 (215.5) | 290.7 (185.7) | 332.5 (217.4) |
| IPAQ-SF (min/week) | 457.5 (1522.0) | 545.0 (1301.3) | 530.0 (1428.8) | 350.0 (690.0) | 420.0 (1022.5) | 400.0 (780.0) | 370.0 (890.0) | 240.0 (415.0) | 315.0 (515.0) |
| GPAQ (min/week) | 765.0 (1603.3) | 810.0 (978.8) | 810.0 (1353.8) | 770.0 (2004.8) | 585.0 (1080.0) | 600.0 (1307.5) | 465.0 (485.0) | 210.0 (765.0) | 345.0 (660.0) |
| Vigorous physical activity | | | | | | | | | |
| RM42 (min/week) | 4.3 (41.0) | 0.2 (12.6) | 0.8 (23.6) | 9.5 (41.2) | 5.2 (34.9) | 6.9 (36.1) | 19.6 (37.2) | 10.2 (37.9) | 13.3 (38.1) |
| IPAQ-SF (min/week) | 250.0 (277.5) | 200.0 (390.0) | 240.0 (322.5) | 210.0 (446.3) | 180.0 (310.0) | 180.0 (330.0) | 270.0 (342.3) | 180.0 (210.0) | 195.0 (300.0) |
| GPAQ (min/week) | 360.0 (490.0) | 240.0 (337.5) | 270.0 (405.0) | 225.0 (390.0) | 242.5 (345.0) | 240.0 (360.0) | 270.0 (607.5 | 255.0 (247.5) | 270.0 (382.5) |
| Moderate to vigorous physical activity | | | | | | | | | |
| RM42 (min/week) | 357.2 (378.8) | 318.1 (232.1) | 324.3 (243.1) | 430.1 (306.9) | 355.2 (291.8) | 390.8 (317.6) | 418.3 (269.4) | 316.6 (232.3) | 377.6 (240.6) |
| IPAQ-SF (min/week) | 757.5 (1665.0) | 795.0 (1413.8) | 795.0 (1582.5) | 485.0 (810.0) | 600.0 (967.0) | 560.0 (937.5) | 510.0 (1330.0) | 420.0 (647.5) | 480.0 (907.5) |
| GPAQ (min/week) | 1260.0 (2081.0) | 930.0 (1462.5) | 960.0 (1620.0) | 882.5 (2515.0) | 810.0 (960.0) | 810.0 (1562.5) | 607.5 (1093.8) | 460.0 (987.5) | 562.5 (877.5) |
| EHIS-PAQ (min/week) | 310.0 (810.0) | 297.5 (301.3) | 297.5 (307.3) | 280.0 (533.8) | 220.0 (445.0) | 280.0 (442.5) | 355.0 (367.5) | 310.0 (220.0) | 327.5 (291.3) |

RM42, accelerometer RM42; IPAQ-SF, International physical activity questionnaire-short form; GPAQ, Global physical activity questionnaire; EHIS-PAQ, European health interview survey–physical activity questionnaire.

$p < 0.001$; GPAQ: $t(178) = -11.6$, $p < 0.001$; EHIS-PAQ: $t(177) = -10.0$, $p < 0.001$) in all fitness groups at the p < 0.001 level. Results for MVPA were statistically significant for IPAQ-SF and GPAQ (IPAQ-SF: $t(175) = 7.48$, $p < 0.001$; GPAQ: $t(162) = 7.54$, $p < 0.001$; EHIS-PAQ: $t(172) = 0.416$, $p = 0.678$). Differences were significant at the $p < 0.001$ level for IPAQ-SF and GPAQ in all fitness groups. There were no statistically significant results for MVPA measured with EHIS-PAQ (high fitness group: $p = 0.901$, intermediate fitness group: $p = 0.313$, low fitness group: $p = 0.109$). MVPA measured by EHIS-PAQ was the only value where the results showed equivalence between accelerometer and PAQ, especially for the high fitness group. The results

**Table 2. Criterion validity between RM42 accelerometer and IPAQ-SF, GPAQ and EHIS-PAQ for three fitness groups.**

| | | | SB | | | MPA | | VPA | | MVPA | | |
|---|---|---|---|---|---|---|---|---|---|---|---|---|
| | | | IPAQ-SF | GPAQ | EHIS-PAQ | IPAQ-SF | GPAQ | IPAQ-SF | GPAQ | IPAQ-SF | GPAQ | EHIS-PAQ |
| Accelerometer | SB | High fitness | .456*** | .410*** | .359** | | | | | | | |
| | | Intermediate fitness | .524*** | .432*** | .601*** | | | | | | | |
| | | Low fitness | .404** | .388** | .317* | | | | | | | |
| | | Total | .468*** | .414*** | .415*** | | | | | | | |
| | MPA | High fitness | | | | .340* | .459*** | | | | | |
| | | Intermediate fitness | | | | .253 | .357* | | | | | |
| | | Low fitness | | | | .205 | .080 | | | | | |
| | | Total | | | | .253** | .292*** | | | | | |
| | VPA | High fitness | | | | | | .049 | .059 | | | |
| | | Intermediate fitness | | | | | | -.094 | .272 | | | |
| | | Low fitness | | | | | | .081 | .192 | | | |
| | | Total | | | | | | .047 | .167 | | | |
| | MVPA | High fitness | | | | | | | | .396*** | .399*** | .407*** |
| | | Intermediate fitness | | | | | | | | .264* | .289* | .485*** |
| | | Low fitness | | | | | | | | .206 | .063 | .491*** |
| | | Total | | | | | | | | .267*** | .237** | .446*** |

* p ≤ 0.01

** ≤ 0.005

*** p ≤ 0.001

MPA, moderate physical activity; VPA, vigorous physical activity; MVPA, moderate to vigorous physical activity; SB, sedentary behaviour; IPAQ-SF, International physical activity questionnaire-short form; GPAQ, Global physical activity questionnaire; EHIS-PAQ, European health interview survey–physical activity questionnaire

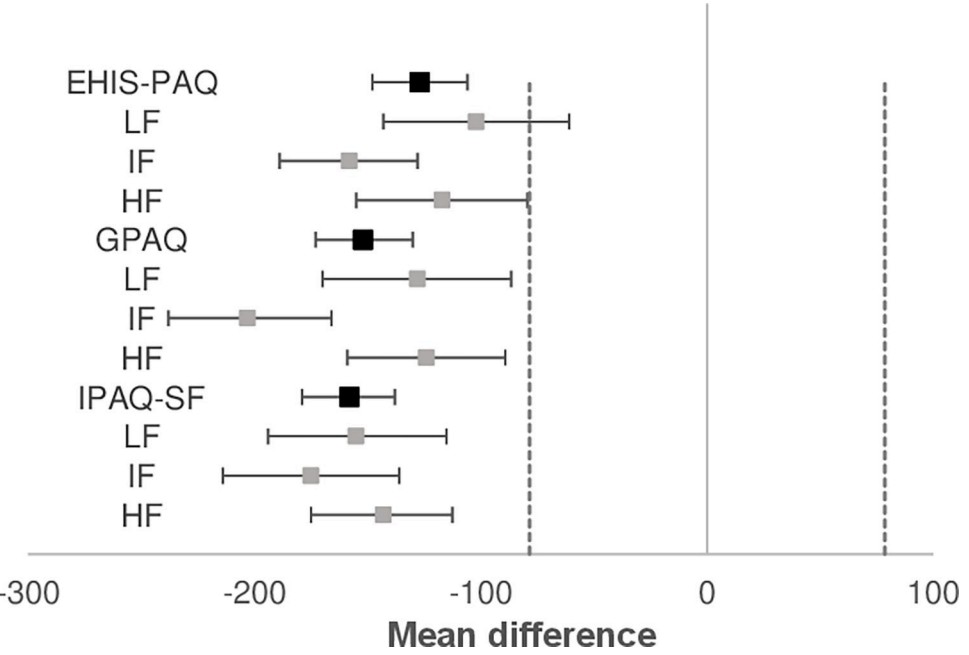

**Fig 1. Bland-Altman plots for the PAQs and UKK RM42 accelerometer for sedentary behaviour (min/day) with 95% limit of agreement.** LF, Low fitness group; IF, Intermediate fitness group; HF, high fitness group; SB, sedentary behaviour.

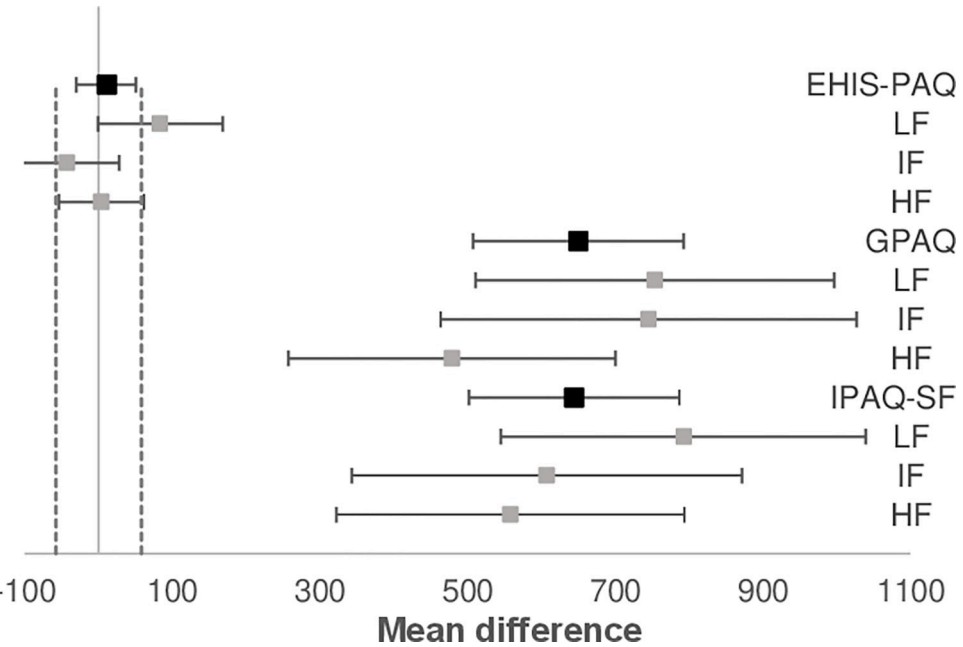

**Fig 2. Bland-Altman plots for the PAQs and UKK RM42 accelerometer for to MVPA (min/week) with 95% limit of agreement.** LF, Low fitness group, IF, Intermediate fitness group, HF, high fitness group, MVPA, moderate to vigorous physical activity.

of IPAQ-SF and GPAQ were not within the equivalence bounds for PA (dotted line), but the results of the high fitness group were closest to the equivalence bounds.

## Discussion

In the present study, we compared the criterion validity of the Slovenian versions of three PAQs popular in Europe (IPAQ-SF, GPAQ and EHIS-PAQ) between differently fit participants. Results showed that self-reported movement behaviour assessed with IPAQ-SF and GPAQ is more comparable to the accelerometer UKK RM42 results for PA and sedentary behaviour in fitter individuals. The same trend was found for EHIS-PAQ, where questions on PA are based on activity descriptions. However, the differences between fitness groups for PA were not significant. In addition, EHIS-PAQ proved to be the most equivalent to the accelerometer assessment of PA among all selected PAQs.

The self-reported PA in our sample was slightly higher compared with other studies in European countries [29, 48]. However, none of the participants included in the study exceeded the maximum daily or weekly value of PA [20, 49]. We hypothesise that Slovenian participants may be more active compared with some other European countries, but similar or higher PA values have been reported in some prior studies. In Hungary, participants reported similar VPA levels when using IPAQ-SF (180 min/week) [50]. In a Lithuanian study, more MPA and MVPA measured with IPAQ-SF was reported (MPA = 490 min/week; MVPA = 600 min/week) [34]. Riviere and colleagues [33] reported higher self-reported PA in the French sample for IPAQ (MPA = 750 min/week, VPA = 880 min/week) and GPAQ (MPA = 900 min/week, VPA = 900 min/week). Difference in self-reported PA between differently fit individuals was previously reported in adolescents, with low-fit participants also over-reporting PA more [51]. Over-reporting of PA and under-reporting of sedentary behaviour is typically present when using PAQs [52, 53], but only one previous study has shown how over-reporting may differ

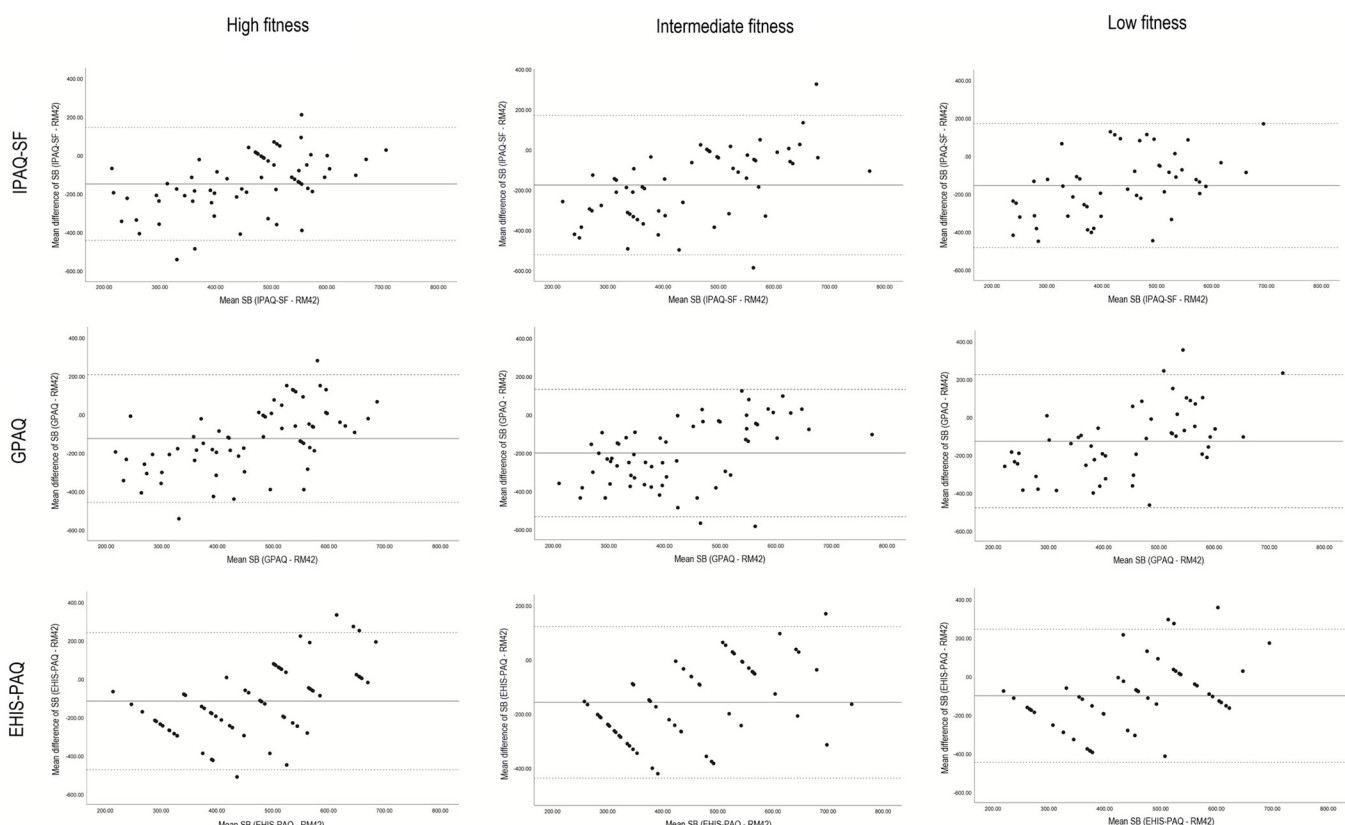

**Fig 3. Observed difference in minutes between UKK RM42 and PAQs for sedentary behaviour in minutes.** Black points represent all data for each PAQ, from bottom to top grey dots represent high, intermediate and low fitness group for each of the PAQs. LF, Low fitness group, IF, Intermediate fitness group, HF, high fitness group.

between different adult fitness groups [23]. This is in line with our findings-the high fitness group overreported MVPA and MPA the least, whereas the low fitness group overreported it the most, compared to the accelerometer results. At the same time, underreporting of sedentary behaviour was lowest in the high fitness group and highest in intermediate fitness group, except at EHIS-PAQ, where the low fitness group underreported the least, when comparing results to accelerometer.

Validity of sedentary behaviour was similar between fitness groups and highest in the intermediate fitness group (Spearman's ρ = 0.432–0.601), although the differences in sedentary behaviour duration were greatest in this group (1111–1415 min/week). The exception to this is the results from EHIS-PAQ, where the data and analysis of this behaviour are moderately different because the data are not reported in exact hours and minutes (as in IPAQ-SF and GPAQ), but rather participants choose from the options offered (e.g., less than 4 hours, 4 to 6 hours, etc.). Overall results for sedentary behaviour validity were higher than European data from the recent meta-analysis of sedentary behaviour questions (weighted mean for criterion validity = 0.23) [54].

The main finding supporting our hypothesis is that the agreement between the PAQs and the accelerometer recordings of the self-reported MPA and MVPA values of IPAQ-SF and GPAQ decrease from the high to the low fitness group (IPAQ-SF MVPA: high fitness group = 0.40, intermediate fitness group = 0.26, low fitness group = 0.21; GPAQ MVPA: high fitness group = 0.40, intermediate fitness group = 0.29, low fitness group = 0.06). In addition, overreporting of MVPA

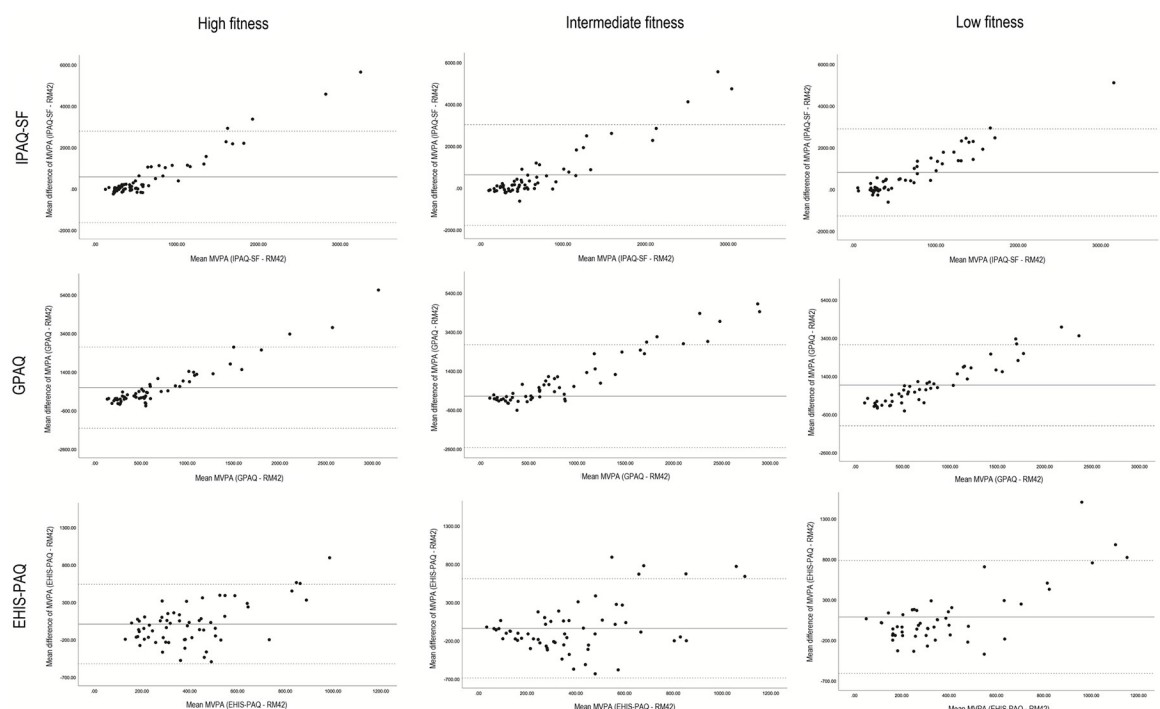

**Fig 4. Observed difference in minutes between UKK RM42 and PAQs for MVPA in minutes.** Black points represent all data for each PAQ, from bottom to top grey dots represent high, intermediate and low fitness group for each of the PAQs. LF, Low fitness group, IF, Intermediate fitness group, HF, high fitness group.

and underreporting of sedentary behaviour compared to accelerometer results were lowest in the high fitness group. In contrast, Shook and colleagues [25] reported higher criterion validity (against Sense Wear Armband) of the IPAQ in unfit participants. Validity was higher for MPA (fit = 0.11, unfit = 0.26) and MVPA (fit = 0.16, unfit = 0.3). In addition, some difference in validity was found for MPA and MVPA compared with other studies. In our study, the validity coefficients were low to very low, however, the results in the high fitness group (MPA: IPAQ-SF = 0.34, GPAQ = 0.46; MVPA: IPAQ-SF = 0.4, GPAQ = 0.4) were among the highest compared to a recent meta-analysis [55]. On the other hand, validity results for the low fitness group were among the lowest reported validity results for IPAQ-SF and GPAQ. For all three PAQs, the agreement between accelerometer results and PAQs self-reported for VPA was very low in all fitness groups in the present study. There were no correlations between self-reported VPA and accelerometer-measured VPA, even in the high fitness group. Since the difference between accelerometer based VPA and self-reported VPA was the biggest in all groups (more than 80% of overreporting), this could influence the poor validity result. In addition, the validity results of VPA were among the lowest reported [55]. The low validity results between accelerometer UKK RM42 and PAQs could be a result of different constructs measured with each of the methods. Accelerometers tend to poorly measure some bipedal activities, such as cycling or skiing, but participants can self-report all those activities with PAQs [11]. Nonetheless, PAQs are subjective measures and depend primarily on individuals retrospective reporting of movement behaviours, specifically participants are least precise when self-reporting sedentary behaviour, where differences between subjective and objective measures are large [56, 57].

The better validity of the fitter participants when using IPAQ-SF and GPAQ can be explained by the assessment items used to determine PA different intensities in these two

instruments. Indeed, the descriptions are based on physical responses to PA (e.g., heavy breathing, increase in heart rate), which are highly subjective and depend primarily on the cardiorespiratory fitness of the individual. Therefore, it is not surprising that more active participants have higher cardiorespiratory fitness and therefore more accurately estimate their PA. Thus, equivalence testing on these two instruments showed statistically significant differences between PAQs and accelerometer assessments for PA and sedentary behaviour. Nevertheless, the results again showed differences between fitness groups, as the high fitness group came closest to the equivalence bounds, but participants in all groups underreported time spent sitting. Similar was found for MVPA, where the difference for IPAQ-SF and GPAQ was significant, and the results were not within the equivalence bounds.

EHIS-PAQ performed the best on the equivalence testing regardless of fitness group. However, although there were no significant differences in validity between the fitness groups, we can notice a trend. The results of the high fitness group were most equivalent to the UKK 42 accelerometer, while the intermediate fitness group tended to underreport and the low fitness group tended to overreport, but the main result was still within the equivalence bound. Since EHIS-PAQ does not measure total PA or PA by intensity, it still gave us the best validity results and the best equivalence compared to the accelerometer. Considering that EHIS-PAQ was developed as a part of the European Health Interview Survey, the design of the questionnaire is different than in other two used PAQs: the intensity of PA was intentionally excluded because participants had a difficulty distinguishing between different intensities of PA [22]. This could be the explanation why we did not find differences between fitness groups when using EHIS-PAQ. The questionnaire also includes recreational activities that are not included in other PAQs and are primarily health enhancing type of PA [58].

## Strengths and limitations

This is one of the first studies to compare the differences between differently fit individuals in terms of their subjective and self-reported PA However, the study has some limitations. First, the study sample was not representative. Because we formed the fitness groups by dividing the participants into terciles, a possible bias in the fitness level of the participants (e.g., participants who were fitter than average) could affect the results of the study. Second, the accelerometer results are dependent on the body placement and metrics used; thus they have limitations assessing some movement behaviours (e.g., swimming, cycling, jumping on trampoline). This should be considered when interpreting results of our study, however validity of accelerometers is still much higher than in PAQs compered to doubly labelled water as a golden standard [13, 59]. Third, field fitness test, i.e., 6 minutes of walking, only assessing and not objectively measure cardiorespiratory fitness was used to determine fitness level. However, this is the popular test in patients and older adults [60] with several advantages: it is simple and can be performed indoors, no equipment is required, and it is not intimidating to participants [41]. Fourth, the MAD algorithm used to analyse accelerometer data in the present study has been validated for bipedal [36]. This could affect the intensity of activities, such as cycling, which may be underestimated, and consequently the volume of VPA measured by the accelerometer may be underestimated. However, similar problems with the measurement of VPA have been highlighted in other studies that used other algorithms for accelerometer data [61, 62].

## Conclusions

The present study showed differences in self-reporting PA between differently fit individuals. The differences in validity of the PAQs among differently fit individuals highlight another dilemma of PAQs. It shows the importance of validating PAQs, not only between nations and

cultures, but also between differently fit individuals. Even though self-report PA by intensity is common in PAQs, our results showed that this type of question is not the most appropriate for all fitness groups. EHIS-PAQ, which does not include PA intensities, performed the best in validity and equivalence testing regardless of fitness group and is therefore the most appropriate PAQ for measuring PA without knowing the fitness level of participants. We believe that future research is needed and would like to emphasise the importance of critically evaluating data collected with PAQs. One contextual piece of information for interpreting PAQ results that can be collected in epidemiological studies and surveillance could be body mass index as a proxy for participants' physical fitness.

## Supporting information

**S1 Data. Accelerometer and PAQs dataset.**
(SAV)

## Acknowledgments

The authors thank the UKK Institute for providing the RM42 accelerometers and for help with data processing. Special thanks to Dr. Saša Đurić for his assistance in data collection, to Antonio Martinko for his advice in the analysis of the equivalence testing analysis and to all school coordinators for their help in organizing the measurements.

## Author Contributions

**Conceptualization:** Paulo Rocha, Gregor Jurak.

**Data curation:** Vedrana Sember, Henri Vähä-Ypyä.

**Formal analysis:** Kaja Meh.

**Funding acquisition:** Paulo Rocha.

**Investigation:** Kaja Meh, Vedrana Sember, Gregor Jurak.

**Methodology:** Kaja Meh, Vedrana Sember.

**Project administration:** Paulo Rocha, Gregor Jurak.

**Software:** Henri Vähä-Ypyä.

**Supervision:** Maroje Sorić, Gregor Jurak.

**Validation:** Kaja Meh.

**Visualization:** Kaja Meh.

**Writing – original draft:** Kaja Meh, Maroje Sorić, Gregor Jurak.

**Writing – review & editing:** Kaja Meh, Vedrana Sember, Maroje Sorić, Henri Vähä-Ypyä, Paulo Rocha, Gregor Jurak.

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
