## [Decision Letter · Decision Letter 0]

22 Feb 2023

PONE-D-22-33819The dilemma of physical activity questionnaires: fitter people are less prone to over reportingPLOS ONE

Dear Dr. Meh,

Thank you for submitting your manuscript to PLOS ONE. After careful consideration, we feel that it has merit but does not fully meet PLOS ONE’s publication criteria as it currently stands. Therefore, we invite you to submit a revised version of the manuscript that addresses the points raised during the review process.

I recommend following the reviewers' comments that are concordat in suggesting major revisions. The manuscript is interesting and I warmly recommend providing a reviewed version for further consideration in PLOS ONE.  

We look forward to receiving your revised manuscript.

Kind regards,

Giulia Squillacioti

Academic Editor

PLOS ONE

Journal Requirements:

“I have read the journal's policy and the authors of this manuscript have the following competing interests: Author Henri Vähä-Ypyä is employee of the UKK Institute which is the main owner of company UKK tervepalvelut that developed RM42 accelerometers.”

Reviewers' comments:

Reviewer's Responses to Questions

**Comments to the Author**

1. Is the manuscript technically sound, and do the data support the conclusions?

Reviewer #1: Yes

Reviewer #2: Yes

2. Has the statistical analysis been performed appropriately and rigorously? 

Reviewer #1: Yes

Reviewer #2: Yes

3. Have the authors made all data underlying the findings in their manuscript fully available?

Reviewer #1: Yes

Reviewer #2: Yes

4. Is the manuscript presented in an intelligible fashion and written in standard English?

Reviewer #1: Yes

Reviewer #2: Yes

5. Review Comments to the Author

Reviewer #1: OVERALL

This article reads well, the presentation is easy to follow, and the data analysis is excellent. Creating three fitness groups to compare PAQ results and accelerometer counts was novel and may help researchers understand PA and movement data by fitness levels.

This reviewer is concerned about how the authors compare the PAQs and the accelerometer results. The two measure different constructs, and it is incorrect to assume the accelerometer is the ‘gold standard’ for establishing movement levels. One cannot assume that the self-report of one’s PA on a PAQ is under-estimated or over-estimated based on accelerometer counts. Studies show the accelerometer depends on movement counts to identify changes in body positions. Many body positions and types of movement are not registered as counts on an accelerometer (e.g., bicycling, economical running, cross-country skiing, skating, skateboarding) but are performed vigorously or at moderate intensities. Perhaps it is better to rephrase the correlations in this article as showing agreement between the PAQ and accelerometer responses and not under-reporting or over-reporting the responses. There is less value judgment of the PAQ responses.

Also, an interesting question is how well the three PAQs correlate for the sedentary, moderate, vigorous, and MVPA responses. This information would inform researchers engaged in surveillance and research about the consistency of results between populations.

SPECIFIC COMMENTS

Line 41-100. The reader needs to understand that accelerometers measure objective movement intensity and PAQ measure subjective recall of PA intensities. The two measures are different constructs, and their differences have been discussed in the literature. Please note the differences in the introduction, so the reader interprets the data with differences between the measures in mind.

Line 102-110. Why did you target adults of 12-14-year-old adolescents to enroll in the study? Can the authors state why this sampling approach was used?

Line 114-116. The meaning of this sentence needs to be clarified.

Line 125. Please identify the language in the PAQ’s were administered. Are there metrics that specify the translation process?

125-135. Please describe the units the questionnaires ask for each item and how each questionnaire was scored in this section or as an appendix to the paper. The reader needs to know the units that the PA is presented and how items are scored. For example, the IPAQ-SF scoring instructions yield scores in MET-min, yet Table 1 presents the scores in min/week. Clarifications are needed to explain how the authors modified the PAQ’s scoring protocols.

Line 171-176. Determination of the participants’ fitness levels is important for this study’s internal validity. More information is needed.

1. Please identify the validity and reliability of the 6-minute walking test and the type of populations the validity and reliability were determined.

2. How was the test scored to identify the fitness groups? Terciles based on the distance covered?

3. What clothing did the participants wear while completing the fitness test?

4. How was the distance covered in the walking test measured?

5. How many participants were measured in the fitness test simultaneously? One at a time or more than one?

6. How many times did each participant complete the test? Once or more than once to familiarize themselves with the test?

Line 181. Please identify the cut points for the distance of the three fitness groups in a parenthesis.

Table 1. This is a very detailed table with much information. However, much of the data for the PAQs are easier to compare if the authors describe how each questionnaire is scored.

Table 2. The presentation for this table is very nice.

Discussion. The discussion is very long and seems repetitive. It should be shortened and focused on explaining plausible reasons for the results.

Line 224 to 232. Because of the different constructs of objective movement recordings vs. recall of PA, is it correct to assume the accelerometer is the ‘gold standard’ of PA? This question is difficult to answer, but one that pits the PAQ as inferior to the accelerometers in assessing PA. PAQs have been used for decades to establish the relationship between PA and health and mortality. At the same time, accelerometers have recently been used for various purposes. The assumption that an accelerometer-recorded movement is superior to the PAQ may not be valid. Perhaps the author can revise the wording in this section to not imply that the PA was not overreported compared to the accelerometer but that the low correlations may represent the measurement of different aspects of the movement. Take bicycling, economical running, cross-country skiing, and skating as MVPA or vigorous activities that register very low counts on an accelerometer. These activities will be classified as low intensity. Hence, one can’t assume the PA is overestimated on a PAQ when the type and performance of an activity fail to register acceleration counts on an accelerometer.

Line 272-277. Is it possible to restate the conclusion of the analyses to state that the measures were similar or not similar instead of underreporting and overreporting one’s PA on the PAQ? Again, such wording implies that an accelerometer is a superior movement-measuring device. This assumption is not the case for all forms of movement, as noted earlier.

Line 288-290, 295-302. How is it possible to state that there is under-reporting of sedentary behavior on a PAQ with the accelerometer that fails to differentiate between low-intensity PA and little-to-no movement?

Line 303. Wouldn’t this initial statement be more correct by stating that there is ‘agreement between the PAQ and the accelerometer recordings’ instead of validity? Such a statement does not imply the accelerometer is the gold standard for PA measurements.

Line 310-313. The low Spearman correlations may result from comparing two uniquely different constructs. Low correlations have been established previously between PAQs and accelerometers, as shown in the article from Craig et al. and others.

Strength and Limitations. This section makes no mention of the limitations of comparing accelerometers and PAQs. This omission is a significant limitation of this article. Estimating fitness levels using the 6-min walk test is a limitation to the validity of the fitness measures.

Line 360-361. Can the author identify the population this sample differed?

Reviewer #2: This manuscript is of interest, especially as the need to quantify physical activity in varying populations grows. While several investigations have looked at the relationship between self-report PA and objective accelerometry, few have looked at the factors which may moderate these relationships. Overall, this study is sound, although limited by the use of the six-minute walk test, and there can be some improvements in the clarity of writing and conclusions made. In particular, the overuse of acronyms makes readability so difficult, it does impact my ability to judge the soundness of the research. I have some specific comments as follows:

Comment 1: Line 112 - the number of participants excluded from the analysis is very high (>50%). Can the authors provide some detail on the reasons for exclusion, particularly where it is due to invalid questionnaire/accelerometer data. Can the authors also clarify what made this data valid and if participants were required to have all three questionnaires completed.

Comment 2: for participants with less than 7 days data, how were the daily average values calculated (e.g. was there always 1 weekend counted)?

Comment 3: Can the authors consider improving the readability of the results. For example, from line 196, the excessive use of abbreviations means these 4 lines are extremely difficult to interpret. The same applies for table 1. I acknowledge that table 1 has a lot of information in it, but it is very hard to compare the measures, groups and intensities measured as they are physically far apart. How could this be improved for the reader?

Comment 4: Given the three surveys ask very similar information, can the authors make comment on the agreement between these measures?

Comment 5: Line 247 - what is a TOST? Please review all acronyms and consider only using when absolutely necessary.

Comment 6: Could the authors have consider using Linear regressions and test for interactions between the self-report tool and fitness for predicting the objective PA measure? The intercepts of the models can be used to determine the significance of any absolute differences.

Comment 7: Could the authors report the overall correlations between the self-report and accelerometers, ignoring fitness. I do note that depending on your response to comment 6 this may not be necessary.

6. PLOS authors have the option to publish the peer review history of their article (what does this mean?). If published, this will include your full peer review and any attached files.

Reviewer #1: No

Reviewer #2: No

---

## [Author Response · Author response to Decision Letter 0]

17 Mar 2023

Academic editor, Plos One

Dear academic editor and reviewers,

On behalf of my co-authors, I would like to thank you for taking your time to read our manuscript and giving us the opportunity to revise and improve our paper. We are grateful for your suggestions and believe they increased the value of our manuscript. You can find our replies to your comments in Italics below.

Journal Requirements:

Thank you for your comment, we checked the manuscript again and made necessary changes to meet the PLOS ONE’s style requirement:

- We changed the font in Headings 1 and 2,

- We revised the citing in the text,

- We renamed the files uploaded with the manuscript,

- We revised the heading of tables and figures,

- We revised the font and dimensions of figures.

“I have read the journal's policy and the authors of this manuscript have the following competing interests: Author Henri Vähä-Ypyä is employee of the UKK Institute which is the main owner of company UKK tervepalvelut that developed RM42 accelerometers.”

There are no restrictions on data sharing, and we confirm it does not alter the adherence to PLOS ONE policies on sharing the data. Our updated Competing interested statement can be found below. 

“I have read the journal's policy and the authors of this manuscript have the following competing interests: Author Henri Vähä-Ypyä is employee of the UKK Institute which is the main owner of company UKK tervepalvelut that developed RM42 accelerometers. This does not alter our adherence to PLOS ONE policies on sharing data and materials.”

We added the captions at the end of the manuscript.

Reviewer #1: OVERALL

This article reads well, the presentation is easy to follow, and the data analysis is excellent. Creating three fitness groups to compare PAQ results and accelerometer counts was novel and may help researchers understand PA and movement data by fitness levels.

This reviewer is concerned about how the authors compare the PAQs and the accelerometer results. The two measure different constructs, and it is incorrect to assume the accelerometer is the ‘gold standard’ for establishing movement levels. One cannot assume that the self-report of one’s PA on a PAQ is under-estimated or over-estimated based on accelerometer counts. Studies show the accelerometer depends on movement counts to identify changes in body positions. Many body positions and types of movement are not registered as counts on an accelerometer (e.g., bicycling, economical running, cross-country skiing, skating, skateboarding) but are performed vigorously or at moderate intensities. Perhaps it is better to rephrase the correlations in this article as showing agreement between the PAQ and accelerometer responses and not under-reporting or over-reporting the responses. There is less value judgment of the PAQ responses.

Also, an interesting question is how well the three PAQs correlate for the sedentary, moderate, vigorous, and MVPA responses. This information would inform researchers engaged in surveillance and research about the consistency of results between populations.

Thank you for taking your time and reading our manuscript. We find your comments valuable and tried to include them in the improved version of the manuscript. We added more information regarding accelerometer and PAQs measurement of movement behavior and tried to point out the differences between the two measures. We do have information about the correlation of movement behavior items between the used PAQs: it was published in a previous paper; however, we also have the information for the data included in present manuscript and can include it as a supporting information see response to comment 4 from Reviewer #2). We considered all your following comments and provided explanation in italics, under each comment.

Meh, K.; Sember, V.; Đurić, S.; Vähä-Ypyä, H.; Rocha, P.; Jurak, G. Reliability and Validity of Slovenian Versions of IPAQ-SF, GPAQ, and EHIS-PAQ for Assessing Physical Activity and Sedentarism of Adults. Int. J. Environ. Res. Public Health 2022, 19, 430. https://doi.org/10.3390/ijerph19010430

SPECIFIC COMMENTS

Line 41-100. The reader needs to understand that accelerometers measure objective movement intensity and PAQ measure subjective recall of PA intensities. The two measures are different constructs, and their differences have been discussed in the literature. Please note the differences in the introduction, so the reader interprets the data with differences between the measures in mind.

Thank you for your comment, we agree that measures differ and wanted to include that information in the introduction, but it was not clear enough. We have added additional information in the lines 52– 72.

Line 102-110. Why did you target adults of 12-14-year-old adolescents to enroll in the study? Can the authors state why this sampling approach was used?

We used a sampling method with which we wanted to ensure as impartial a sample as possible. By including primary schools and their students, we were able to invite different individuals from different backgrounds to the research. 

Line 114-116. The meaning of this sentence needs to be clarified.

Thank you for your comment, we added some details and hopefully improved the meaning of the sentence (lines 128 – 129).

Line 125. Please identify the language in the PAQ’s were administered. Are there metrics that specify the translation process?

We provided a more detailed description of the translation process in lines 156 – 159. We used the forward-backward translation process, recommended by the World Health Organization and included independent translators to complete this process. During the translation we did not find any major discrepancies between the translators.

125-135. Please describe the units the questionnaires ask for each item and how each questionnaire was scored in this section or as an appendix to the paper. The reader needs to know the units that the PA is presented and how items are scored. For example, the IPAQ-SF scoring instructions yield scores in MET-min, yet Table 1 presents the scores in min/week. Clarifications are needed to explain how the authors modified the PAQ’s scoring protocols.

We added additional information about units from each of the questionnaires in the lines 139 - 145). All three questionnaires are collecting weekly physical activity data and daily sedentary behavior data. The MET values are later on calculated from the weekly self-reported physical activity. As MET values differ between PAQs and between accelerometer (e.g. walking in IPAQ-SF = 3.3 MET, transport in GPAQ = 4 MET, MPA in accelerometer = 3 MET, EHIS-PAQ does not include MET calculations). We do believe that there should be a more unified approach in MET calculation, which would enable comparison of MET values between different PAQs and different methods. Since this approach is not unified between the methods used in our paper, we decided to compare time spent in different movement behaviors (e.g. MPA, VPA, SB), as participants should be able to self-assess it similarly. In similar PAQ validation studies, researchers also compared times spent in different movement behaviors. 

Line 171-176. Determination of the participants’ fitness levels is important for this study’s internal validity. More information is needed.

1. Please identify the validity and reliability of the 6-minute walking test and the type of populations the validity and reliability were determined.

2. How was the test scored to identify the fitness groups? Terciles based on the distance covered?

3. What clothing did the participants wear while completing the fitness test?

4. How was the distance covered in the walking test measured?

5. How many participants were measured in the fitness test simultaneously? One at a time or more than one?

6. How many times did each participant complete the test? Once or more than once to familiarize themselves with the test?

Thank you for the comments and additional questions that helped us improve this part of the methodology. In the lines 195-210 we improved the text and described the test and protocol in more detail. We are writing additional comments below:

1. Even though 6-minute walk test is usually used in clinical populations; a few studies tested its measurement characteristics in healthy participants. It is a suitable measure of cardiorespiratory fitness in children and youth 

(Li, A. M., Yin, J., Yu, C. C. W., Tsang, T., So, H. K., Wong, E., ... & Sung, R. (2005). The six-minute walk test in healthy children: reliability and validity. European Respiratory Journal, 25(6), 1057-1060., 

Castro-Piñero J, Artero EG, España-Romero V, Ortega FB, Sjöström M, Suni J, Ruiz JR. Criterion-related validity of field-based fitness tests in youth: a systematic review. Br J Sports Med. 2010 Oct;44(13):934-43. doi: 10.1136/bjsm.2009.058321. Epub 2009 Apr 12. PMID: 19364756.) 

and in adults

(Burr JF, Bredin SS, Faktor MD, Warburton DE. The 6-minute walk test as a predictor of objectively measured aerobic fitness in healthy working-aged adults. Phys Sportsmed. 2011 May;39(2):133-9. doi: 10.3810/psm.2011.05.1904. PMID: 21673494., 

Mänttäri, A., Suni, J., Sievänen, H., Husu, P., Vähä-Ypyä, H., Valkeinen, H., Tokola, K. and Vasankari, T. (2018), Six-minute walk test: a tool for predicting maximal aerobic power (VO2 max) in healthy adults. Clin Physiol Funct Imaging, 38: 1038-1045. https://doi.org/10.1111/cpf.12525)

2. The 6-minute walk distance was used as a score of 6-minute walk test. We cleared that up in line 208.

3. During the fitness testing participants were wearing a sport wear (T-shirt and shorts, leggings or tracksuit and athletic footwear). We added this information in lines 202 - 204.

4. We marked the track with cones 5 meters apart. The researcher counted the number of full laps each participant covered in 6 minutes (from the starting cone) and then added the remaining distance to the last cone they reached. We added this information in lines 207 - 209.

5. Maximum of 4 participants were measured at once, if there were enough researchers to keep track of the distance covered. We added this information in the lines 209 - 210.

6. Each participant completed the test once. If they would have completed it multiple times, tiredness could have influenced their performance and 6-minute walk distance. We added this information in the lines 209 - 210.

Line 181. Please identify the cut points for the distance of the three fitness groups in a parenthesis.

We added this information separated by gender in lines 215 - 218, it should be noted that distance was also calculated for the 4 age groups described in the same paragraph.

Table 1. This is a very detailed table with much information. However, much of the data for the PAQs are easier to compare if the authors describe how each questionnaire is scored.

We described the scoring of the questionnaires in the Methods section of the paper in lines 150 - 155. Furthermore, we made some changes to Table 1 to make it more readable.

Table 2. The presentation for this table is very nice.

Thank you for your comment. 

Discussion. The discussion is very long and seems repetitive. It should be shortened and focused on explaining plausible reasons for the results.

Line 224 to 232. Because of the different constructs of objective movement recordings vs. recall of PA, is it correct to assume the accelerometer is the ‘gold standard’ of PA? This question is difficult to answer, but one that pits the PAQ as inferior to the accelerometers in assessing PA. PAQs have been used for decades to establish the relationship between PA and health and mortality. At the same time, accelerometers have recently been used for various purposes. The assumption that an accelerometer-recorded movement is superior to the PAQ may not be valid. Perhaps the author can revise the wording in this section to not imply that the PA was not overreported compared to the accelerometer but that the low correlations may represent the measurement of different aspects of the movement. Take bicycling, economical running, cross-country skiing, and skating as MVPA or vigorous activities that register very low counts on an accelerometer. These activities will be classified as low intensity. Hence, one can’t assume the PA is overestimated on a PAQ when the type and performance of an activity fail to register acceleration counts on an accelerometer.

Thank you for your valuable comment. We changed the wording in this part and did not use the words over and under estimation (lines 268 – 271 and 273 – 276). We changed the wording in other parts of the discussion in line with your comment and suggestion.

Line 272-277. Is it possible to restate the conclusion of the analyses to state that the measures were similar or not similar instead of underreporting and overreporting one’s PA on the PAQ? Again, such wording implies that an accelerometer is a superior movement-measuring device. This assumption is not the case for all forms of movement, as noted earlier.

We changed this part of the text and explained in more detail differences in measurement of PA between accelerometer and PAQs in lines 319 – 323.

Line 288-290, 295-302. How is it possible to state that there is under-reporting of sedentary behavior on a PAQ with the accelerometer that fails to differentiate between low-intensity PA and little-to-no movement?

The accelerometer used in present study did not report difficulties in differentiating between low-intensity physical activity and sedentary behavior. The MAD method used with the UKK RM42 accelerometer can differentiate between different intensity levels of activity and sedentary behavior 1,2. Similar results were obtained in other studies, where the term under-reporting was also used 3,4,5. Nevertheless, we decided to change the wording to compared to accelerometer to make it clearer that the difference is only present when comparing PAQ results to the UKK RM42 accelerometer.

1 Vähä-Ypyä, H., Vasankari, T., Husu, P., Suni, J. and Sievänen, H. (2015), A universal, accurate intensity-based classification of different physical activities using raw data of accelerometer. Clin Physiol Funct Imaging, 35: 64-70. https://doi.org/10.1111/cpf.12127

2 Vähä-Ypyä, H, Husu, P, Suni, J, Vasankari, T, Sievänen, H. Reliable recognition of lying, sitting, and standing with a hip-worn accelerometer. Scand J Med Sci Sports. 2018; 28: 1092- 1102. https://doi.org/10.1111/sms.13017

3 Bakker, E.A., Hartman, Y.A.W., Hopman, M.T.E. et al. Validity and reliability of subjective methods to assess sedentary behaviour in adults: a systematic review and meta-analysis. Int J Behav Nutr Phys Act 17, 75 (2020). https://doi.org/10.1186/s12966-020-00972-1

4 Healy GN, Clark BK, Winkler EA, Gardiner PA, Brown WJ, Matthews CE. Measurement of adults' sedentary time in population-based studies. Am J Prev Med. 2011 Aug;41(2):216-27. doi: 10.1016/j.amepre.2011.05.005. PMID: 21767730; PMCID: PMC3179387.

5 Prince, S.A., Cardilli, L., Reed, J.L. et al. A comparison of self-reported and device measured sedentary behaviour in adults: a systematic review and meta-analysis. Int J Behav Nutr Phys Act 17, 31 (2020). https://doi.org/10.1186/s12966-020-00938-3

Line 303. Wouldn’t this initial statement be more correct by stating that there is ‘agreement between the PAQ and the accelerometer recordings’ instead of validity? Such a statement does not imply the accelerometer is the gold standard for PA measurements.

Thank you for your suggestion, we found it very helpful and changed the sentence in lines 352– 353.

Line 310-313. The low Spearman correlations may result from comparing two uniquely different constructs. Low correlations have been established previously between PAQs and accelerometers, as shown in the article from Craig et al. and others.

Low correlations are usually reported when comparing physical activity questionnaires and accelerometers and our study did not differ in that aspect. We do agree that low correlations could be a result of different constructs and included that in the manuscript.

Strength and Limitations. This section makes no mention of the limitations of comparing accelerometers and PAQs. This omission is a significant limitation of this article. Estimating fitness levels using the 6-min walk test is a limitation to the validity of the fitness measures.

We mentioned the use of 6-minute walk test as a limitation in lines 440 – 445. We have added the limitation of comparing accelerometer with PAQs in lines 434 – 440.

Line 360-361. Can the author identify the population this sample differed?

We wanted to include a sample of healthy adults from Slovenia, but the sample included in the study is not representative of this population, as age distribution and BMI distribution differ from the populations.

Reviewer #2: This manuscript is of interest, especially as the need to quantify physical activity in varying populations grows. While several investigations have looked at the relationship between self-report PA and objective accelerometry, few have looked at the factors which may moderate these relationships. Overall, this study is sound, although limited by the use of the six-minute walk test, and there can be some improvements in the clarity of writing and conclusions made. In particular, the overuse of acronyms makes readability so difficult, it does impact my ability to judge the soundness of the research. I have some specific comments as follows:

Thank you for taking your time and reading our manuscript. We find your comments valuable and tried to include them in the improved version of the manuscript. We tried to improve the readability of the manuscript and lowered the number of abbreviations used.

Comment 1: Line 112 - the number of participants excluded from the analysis is very high (>50%). Can the authors provide some detail on the reasons for exclusion, particularly where it is due to invalid questionnaire/accelerometer data. Can the authors also clarify what made this data valid and if participants were required to have all three questionnaires completed.

Most of the participants were excluded due to the missing fitness data. As in all studies, participation was voluntary and some participants did not compete this part of the study, due to different reasons (illness, did not bring their sportswear, did not want to perform the test). The valid accelerometer data is described in the Objective measures section. Valid questionnaires data was completed each questionnaire, so there were no missing data and included participants completed all three questionnaires. We added this additional information in lines 128 - 129.

Comment 2: for participants with less than 7 days data, how were the daily average values calculated (e.g. was there always 1 weekend counted)?

As the inclusion criteria dictated, only participants with 4 valid days of data were included in the study. But as our data shows, participants included in the study only had one missing day or less. Therefore, there was only one weekend or weekday missing. For these participants average values were calculated for the 6 days of reported data.

Comment 3: Can the authors consider improving the readability of the results. For example, from line 196, the excessive use of abbreviations means these 4 lines are extremely difficult to interpret. The same applies for table 1. I acknowledge that table 1 has a lot of information in it, but it is very hard to compare the measures, groups and intensities measured as they are physically far apart. How could this be improved for the reader?

We have changed Table 1 accordingly: we grouped together the intensities of physical activity and deleted some of the abbreviations. Hopefully it makes it easier to read and compare results.

Comment 4: Given the three surveys ask very similar information, can the authors make comment on the agreement between these measures?

There was a moderate agreement between questionnaires, the best results were for sedentary behavior and for moderate to vigorous physical activity (we added the table with results below). We did not include the results of the agreement between questionnaires in the paper, because we decided to include criterion validity in the present study and other measures (equivalence testing and Bland-Altman plots). Therefore, agreement between the questionnaires would take up too much space in the paper and was excluded from the manuscript. If you find this information crucial and interesting, we could add it as Supporting information.

Table 1. Agreement between IPAQ-SF, GPAQ and EHIS-PAQ.

 IPAQ-SF EHIS-PAQ

 SB MPA VPA Walk MVPA SB Walk Cycle MV Aerobic

Recreational

Activity

GPAQ SB 1 .890*** .828*** 

 2 .765*** .720*** 

 3 .788*** .706*** 

 Work MPA 1 .489** 

 2 .409* 

 3 .601** 

 Work VPA 1 .908*** 

 2 .700* 

 3 .483 

 Leisure MPA 1 .244 

 2 .499** 

 3 .189 

 Leisure VPA 1 .651*** 

 2 .042 

 3 .502** 

 Transport 1 .486** .373* .315 

 2 .415* .708*** .508** 

 3 .671*** .689*** .311 

 MPA 1 .375* 

 2 .605*** 

 3 .468** 

 VPA 1 .801*** 

 2 .202 

 3 .597*** 

 MVPA 1 .665*** .434***

 2 .576** .472***

 3 .597*** .363**

EHIS-PAQ SB 1 .707*** 

 2 .841*** 

 3 .721*** 

 Walk 1 .487*** 

 2 .368** 

 3 .540*** 

 MV Aerobic

Recreational

Activity 1 .258 

 2 .452** 

 3 .375* 

* p ≤ 0.05; ** ≤ 0.01; p ≤ 0.001

Notes: 3 = low fitness group; 2 = intermediate fitness group; 1 = high fitness group; VPA = vigorous physical activity; MPA = moderate physical activity; MVPA = moderate to vigorous physical activity; SB = sedentary behaviour; IPAQ = International physical activity; GPAQ = Global physical activity; EHIS-PAQ = European health interview survey – physical activity questionnaire.

Comment 5: Line 247 - what is a TOST? Please review all acronyms and consider only using when absolutely necessary.

TOST is a test of equivalence, that we used in present study. The TOST stands for two one-sided tests. We changed the abbreviation in the text in line 293 - 294.

Comment 6: Could the authors have consider using Linear regressions and test for interactions between the self-report tool and fitness for predicting the objective PA measure? The intercepts of the models can be used to determine the significance of any absolute differences.

This is a very valuable comment and a great idea for future research. In present study we wanted to explore the differences in self-reporting of the physical activity and sedentary behavior, therefore we did not use the linear regression. But we find your comment and idea a great opportunity for future work in the field of physical activity measurement. 

Comment 7: Could the authors report the overall correlations between the self-report and accelerometers, ignoring fitness. I do note that depending on your response to comment 6 this may not be necessary.

We added the overall correlation coefficients in Table 2 for sedentary behavior, MPA, VPA and MVPA.

---

## [Decision Letter · Decision Letter 1]

24 Apr 2023

The dilemma of physical activity questionnaires: fitter people are less prone to over reporting

PONE-D-22-33819R1

Dear Dr. Meh,

We’re pleased to inform you that your manuscript has been judged scientifically suitable for publication and will be formally accepted for publication once it meets all outstanding technical requirements.

Kind regards,

Giulia Squillacioti

Academic Editor

PLOS ONE

Additional Editor Comments (optional):

Reviewers' comments:

Reviewer's Responses to Questions

**Comments to the Author**

1. If the authors have adequately addressed your comments raised in a previous round of review and you feel that this manuscript is now acceptable for publication, you may indicate that here to bypass the “Comments to the Author” section, enter your conflict of interest statement in the “Confidential to Editor” section, and submit your "Accept" recommendation.

Reviewer #1: All comments have been addressed

Reviewer #2: All comments have been addressed

2. Is the manuscript technically sound, and do the data support the conclusions?

Reviewer #1: Yes

Reviewer #2: Yes

3. Has the statistical analysis been performed appropriately and rigorously? 

Reviewer #1: Yes

Reviewer #2: Yes

4. Have the authors made all data underlying the findings in their manuscript fully available?

Reviewer #1: Yes

Reviewer #2: Yes

5. Is the manuscript presented in an intelligible fashion and written in standard English?

Reviewer #1: Yes

Reviewer #2: Yes

6. Review Comments to the Author

Reviewer #1: Thank you for editing your paper based on the reviewer’s comments. You did a good job. Two more things will be helpful to the reader. First, it would be helpful to have the abbreviations defined for the questionnaires in the abstract. Second, in the strengths and limitations section, addition of a comment that accelerometers and PAQs measure different constructs, hence, lower correlations between the two measures deemed significant are expected.

Reviewer #2: Thank you for addressing my comments, particularly the work in improving the clarity of the writing and use of accronyms. All the best.

7. PLOS authors have the option to publish the peer review history of their article (what does this mean?). If published, this will include your full peer review and any attached files.

Reviewer #1: No

Reviewer #2: No

---

## [Editor Report · Acceptance letter]

12 May 2023

PONE-D-22-33819R1 

The dilemma of physical activity questionnaires: fitter people are less prone to over reporting 

Dear Dr. Meh:

I'm pleased to inform you that your manuscript has been deemed suitable for publication in PLOS ONE. Congratulations! Your manuscript is now with our production department. 

Kind regards, 

on behalf of

Dr. Giulia Squillacioti 

Academic Editor

PLOS ONE